# Lampenflora in a Show Cave in the Great Basin Is Distinct from Communities on Naturally Lit Rock Surfaces in Nearby Wild Caves

**DOI:** 10.3390/microorganisms9061188

**Published:** 2021-05-31

**Authors:** Jake Burgoyne, Robin Crepeau, Jacob Jensen, Hayden Smith, Gretchen Baker, Steven D. Leavitt

**Affiliations:** 1Department of Plant and Wildlife Sciences, Brigham Young University, Provo, UT 84602, USA; jakeburgoyne96@gmail.com; 2Department of Biology, Brigham Young University, Provo, UT 84602, USA; crepeaurobin@gmail.com (R.C.); jnjensen88@gmail.com (J.J.); haydenbsmith813@gmail.com (H.S.); 3Great Basin National Park, National Park Service, Baker, NV 89311, USA; Gretchen_Baker@nps.gov; 4Department of Biology & M.L. Bean Museum of Life Science, Brigham Young University, Provo, UT 84602, USA

**Keywords:** algae, amplicon sequencing, bacteria, biofilm, environmental sampling, diatoms, fungi, Great Basin, high-throughput sequencing, metabarcoding

## Abstract

In show caves, artificial lighting is intended to illuminate striking cave formations for visitors. However, artificial lighting also promotes the growth of novel and diverse biofilm communities, termed lampenflora, that obtain their energy from these artificial light sources. Lampenflora, which generally consist of cyanobacteria, algae, diatoms, and bryophytes, discolor formations and introduce novel ecological interactions in cave ecosystems. The source of lampenflora community members and patterns of diversity have generally been understudied mainly due to technological limitations. In this study, we investigate whether members of lampenflora communities in an iconic show cave—Lehman Caves—in Great Basin National Park (GRBA) in the western United States also occur in nearby unlit and rarely visited caves. Using a high-throughput environmental DNA metabarcoding approach targeting three loci—the ITS2 (fungi), a fragment of the 16S (bacteria), and a fragment of 23S (photosynthetic bacteria and eukaryotes)—we characterized diversity of lampenflora communities occurring near artificial light sources in Lehman Caves and rock surfaces near the entrances of seven nearby “wild” caves. Most caves supported diverse and distinct microbial-dominated communities, with little overlap in community members among caves. The lampenflora communities in the show cave were distinct, and generally less diverse, from those occurring in nearby unlit caves. Our results suggest an unidentified source for a significant proportion of lampenflora community members in Lehman Caves, with the majority of community members not found in nearby wild caves. Whether the unique members of the lampenflora communities in Lehman Caves are related to distinct abiotic conditions, increased human visitation, or other factors remains unknown. These results provide a valuable framework for future research exploring lampenflora community assemblies in show caves, in addition to a broad perspective into the range of microbial and lampenflora community members in GRBA. By more fully characterizing these communities, we can better monitor the establishment of lampenflora and design effective strategies for their management and removal.

## 1. Introduction

Caves represent unique habitats that harbor biological communities that are often distinct from those occurring nearby on the surface [1]. However, cave ecosystems are not entirely isolated from surface habitats. Microbes may be transported from a potential source and into cave systems through dust deposition by air currents and vertical and horizontal fluid flow [1,2,3]. Generally, cave habitats in temperate regions are more climatically stable than the surface, characterized by low positive temperatures, high relative humidity, and lack of light [4]. Natural caves connected to the surface have three major habitat zones based on light penetration and intensity: entrance, twilight, and dark zone [1]. In cave systems, habitats near entrances and other openings are a transition area with moderated daily and seasonal fluctuations. Furthermore, low levels of photon fluxes near the cave entrances allow photoautotrophic communities to develop, which cannot survive in the dark zone of caves. Deep within this dark zone of caves, organic matter from the outside or chemoautotrophs provide the energetic foundation for the development of simple communities. In many show caves, permanent artificial lighting may facilitate the development of unique photoautotrophic communities comprised of cyanobacteria, algae, mosses, and ferns—lampenflora—deep within the dark zone [5] (Figure 1). This novel niche provides a fascinating model for investigating the origin and establishment of microbial-dominated communities in relatively simple cave ecosystems.

Opportunistic lampenflora communities around artificial lighting in developed underground environments create novel ecological networks that otherwise would not exist [6,7,8,9]. Lampenflora communities also cause aesthetic changes, physical biodeterioration, e.g., increased porosity, exfoliation, and structural changes, and chemical biodeterioration (e.g., dissolution of speleothems, and alteration and precipitation of mineral mixtures) [10]. The result, relative to natural photoautotrophic communities near cave entrances, is lampenflora communities that are generally less diverse, with the potential development of a resilient niche of biofilm that is more difficult to manage in show caves [5,10]. Traditionally, these lampenflora communities have been characterized largely based on microscopy- and culture-based methods [11,12,13]. With the advent of high-throughput sequencing and DNA metabarcoding approaches, microbial and lampenflora diversity in caves systems can be more robustly characterized, facilitating new approaches to investigate community establishment and structure [14,15,16]. However, the use of high-throughput sequencing to characterize lampenflora communities is still relatively new, lacking well-established sampling approaches, amplification and sequencing methodologies, and genetic reference databases for sequence comparisons.

Show caves are iconic features in some National Parks in the United States, including Great Basin National Park (GRBA) in Nevada, USA. In the National Park System, several show caves are easily accessible with artificially lit cave features, attracting hundreds of thousands of visitors annually. More recently, issues relating to lampenflora growth have become a major concern for cave management [15,17]. Artificial lighting was first installed in Lehman Caves in GRBA in 1941, with a new incandescent lighting system installed in 1977. In 2006, incandescent bulbs were changed to light emitting diodes (LED) systems to reduce the amount of heat generated from each light [17]. Biodiversity in Lehman Caves has received considerable attention, documenting distinct communities throughout the cave, and preliminary insight into lampenflora communities [17,18,19]. This show cave receives considerable foot traffic with ca. 33,000 human visitors annually, which also likely has an impact on the cave’s ecosystem, e.g., [20]. Regular treatments of diluted sodium hypochlorite are used to attempt to control lampenflora growth near artificial light sources in Lehman Caves, although these treatments do not fully eliminate lampenflora communities.

The connection between the communities near the entrances and twilight zones of wild caves—caves with infrequent human visitation and without artificial lighting—with lampenflora communities in nearby show caves is not well understood [1,4,21]. Microbial and lampenflora communities on rock walls and speleothems (depositional formations, e.g., stalactite or stalagmite) are often distinct from communities found on the surface [22,23], and cave soils often harbor bacteria that are not found in nearby surface soils [24,25]. However, if lit areas in wild caves are important source populations for artificial lampenflora communities are not known. Multiple “wild” caves are found throughout GRBA, each with unique biological and geological properties [19]. We hypothesize that lampenflora communities in Lehman Caves reflect diversity from nearby wild caves. In this study, our overarching objective is to assess if lampenflora community members in an artificially lit show cave, Lehman Caves, in GRBA are a subset of biofilm communities occurring in nearby wild caves. To test this hypothesis, we aim to (i) document diversity of photoautotrophs, non-photosynthetic bacteria, and fungi in lampenflora communities in Lehman Caves and those cave surfaces in eight additional nearby wild caves using high-throughput environmental DNA metabarcoding and (ii) compare community composition and similarity among cave samples.

## 2. Materials and Methods

### 2.1. Cave Biofilm Sampling

Biofilm samples in Great Basin National Park, Nevada, USA, were collected from naturally lit marble, limestone, and dolomite surfaces within and near the entrances of eight “wild” caves distributed across three drainages, in addition to artificially lit biofilm communities in Lehman Caves (Table 1). Biofilm samples were obtained from eight wild caves by swabbing rock surfaces with a sterile, disposable toothbrush. Cave entrances were visually assessed to identify surfaces with any degree of indirect exposure to natural light, sampling approximately 10–20 m^2^ of rock surfaces. We attempted to sample the full range of biofilm communities, with particular emphasis on visible microbial communities and diverse microhabitats, in addition to general, broad sampling. Lampenflora communities in Lehman Caves are periodically treated with 10% sodium hypochlorite (NaClO/bleach) solution to control the growth of lampenflora associated with artificial lighting [26]. This treatment does not eliminate lampenflora, and ongoing treatments occur ca. every six months. For Lehman Caves, samples were collected at two intervals, on 31 May 2019 (ca. four weeks following a bleach treatment) and on 15 November 2019, with no bleach treatment between the two sampling times. For the initial sampling at Lehman Caves, ten distinct artificially lit areas were sampled independently; and five of the areas were subsequently sampled in November 2019. We attempted to sample the full range of lampenflora communities near artificial light sources, which average 800 lumens per light, targeting visible communities and diverse microhabitats, in addition to general, broad sampling. Samples were collected from approximately 1–3 m^2^ of rock surfaces at each site. Samples from the wild caves were collected on 15 November, with the exception of Ice Cave, which was sampled on 31 May 2019. Immediately after sampling, sample brushes were placed directly into 50 mL centrifuge tubes, placed in sterilized, cave-specific bags, and subsequently kept on dry ice (−78 °C) before total DNA extraction within 24 h.

### 2.2. DNA Extraction, Amplification, and Sequencing

DNA was extracted from each cave sample independently using the ZR Fungal/Bacterial DNA Miniprep Kit (Zymo Research, Orange County, CA, USA) and following the manufacturer’s recommendations. Subsamples from DNA extractions of Lehman Caves’ samples were subsequently pooled into two composite samples—“LEHM 1”, comprised of the ten samples from May 2019, and “LEHM 2”, comprised of the five samples from November 2019. To amplify photosynthetic members of the biofilm communities, including artificially lit lampenflora, we used primers p23SrV_f1 (GGACAGAAAGACCCTATGAA) and p23SrV_r1 (TCAGCCTGTTATCCCTAGAG), which flank Domain V of the 23S plastid rRNA gene that is present in cyanobacteria and the plastids of photosynthetic eukaryotes [27]. For non-photosynthetic bacteria, a fragment of 16S rDNA was amplified using primers 799f (AACMGGATTAGATACCCKG) [28] and 1100R (AGGGTTGCGCTCGTTG) [29]. For the fungal portion of the biofilm communities, we amplified the ITS2 region of the internal transcribed spacer region, the formal barcoding marker for fungi, using primers ITS-3 (GCATCGATGAAGAACGCAGC) [30] with ITS4 (TCCTCCGCTTATTGATATGC) [31]. Amplifications were performed and prepared for sequencing at RTL Genomics (Lubbock, TX, USA). Amplification products were pooled equimolar, and each pool was size selected in two rounds using SPRIselect (BeckmanCoulter, Indianapolis, IN, USA) in a 0.7 ratio for both rounds. Size selected pools were then quantified using the Qubit 2.0 fluorometer (Life Technologies, Carlsbad, CA, USA) and loaded into an Illumina MiSeq (Illumina, Inc. San Diego, CA, USA) 2 × 300 flow cell at 10 pM and sequenced at RTLGenomics.

### 2.3. Bioinformatic Pipelines

The sequencing reads obtained for the partial 23S plastid rRNA and bacterial 16S markers were processed using the FROGs pipeline [32]. Operational taxonomic units (OTUs) were inferred using sequence clustering performed using Swarm and a cutoff value of 97% similarity for all datasets (16S, 23S, and ITS2) [33]. Chimeric sequences were detected with the VSEARCH algorithm, by the de novo UCHIME method [34] and were removed. A filtering tool was used to remove OTUs, which had a read number abundance of less than 0.005% of all reads. For 16S and 23S data, OTU taxonomy was assigned with the RDP Classifier [35] and the 16S rRNA Silva database [36]. Initial taxonomic assignments for the 23S OTUs were largely ambiguous (see Section 3), and 23S taxonomic assignments were further refined using a phylogenetic approach to identify major monophyletic clades and BLAST was used to infer the taxonomy of those clades. Seed sequences from the inferred 23S OTUs were aligned using the program MAFFT v. 7 [37,38]. We implemented the G-INS-i alignment algorithm and “20PAM/K = 2” scoring matrix, with an offset value of 0.2, and the remaining parameters were set to default values. Subsequently, ambiguous regions within the alignment were removed using Gblocks v. 0.91b [39] implementing default parameters. The phylogeny of the 23S fragment was reconstructed using maximum likelihood as implemented in IQ-TREE v2 [40], with 1000 ultra-fast bootstrap replicates [41] to assess nodal support, using the best-fit substitution model as predicted by ModelFinder [42].

The fungal ITS2 reads were processed using PIPITS v. 2.3 [43]. PIPITS provides a straightforward pipeline and has been shown to outperform a number of other commonly used methods and offers a good balance between speed, taxonomic accuracy (including support for ITS Extractor), and technical quality [44]. Using PIPITS, sequences were assembled with VSEARCH and quality-filtering was undertaken with fastx through the PIPITS command pispino_createreadpairslist. The ITSx was executed through the PIPITS command pipits_funits. Chimera filtering and clustering were undertaken using VSEARCH through the PIPITS command pipits_process.

### 2.4. Community Analyses

All community analyses were performed using the package Primer v. 7 [45] based on OTUs inferred from the partial 23S, partial 16S, and ITS2 short read data and using the number of reads per OTU: fungi (OTUs inferred from ITS2 reads), bacteria (16S OTUs), and photosynthesizing microbes, bryophytes, and vascular plants (23S OTUs). The 23S lampenflora OTUs were further divided into four separate categories—algae, diatoms, bryophytes, and cyanobacteria—for a more nuanced perspective into lampenflora community relationships. Inadvertent sampling of large clusters of microbes where only a few species may dominate the biofilm community can give a biased perspective of the communities as a whole [46]. To reduce the impact skewed species counts due to the sampling strategy and other outliers, we used dispersion weighting in PRIMER. Because the total species abundance from each site were highly variable, the data were normalized by the species count from each cave into proportions to facilitate more accurate comparisons among cave communities [47]. The normalized data was then log(x + 1) transformed.

Ordination and clustering analyses of samples were run at the OTU level based on Bray–Curtis dissimilarity. We used non-metric multidimensional scaling (NMDS), with 1000 random restarts, to evaluate differences in community composition [48]. We also applied an analysis of similarities [49] on the same matrix, with 999 permutations to test a null hypothesis of no community difference between the show caves and the wild caves. Another ANOSIM test was run comparing the sites based on their respective watersheds, as well as one based on bat presences within in the caves. The Bray–Curtis matrices were also used for hierarchical agglomerative clustering (CLUSTER), so that group average dendrograms could be displayed and a visual analysis of community similarity could occur. Similarity profile analysis (SIMPROF) was used to test for statistical significance and multivariate structure within the clusters formed [50].

## 3. Results

### 3.1. High-Throughput Environmental DNA Sequencing and OTU Clustering

Short reads generated for this study were deposited in NCBI’s Short Read Archive under BioProject accession number PRJNA731535. All DNA amplification attempts for the DNA extraction from the Snake Creek Cave sample failed, and this cave was not included in all analyses and comparisons.

A summary of OTUs per sample is reported in Table 2, with completed OTU tables and FROGS and PIPITS results reported in Appendix A. Overall OTU counts ranged from 254 OTUs in “LEHM 2” to 986 OTUs in Fox Skull Cave (Table 2). The highest OTU diversity inferred from the partial 23S reads—photoautotrophs—was found in Catamount Cave (120 OTUs) and the lowest in Lower Pictograph Cave (10 OTUs). Fungal OTUs inferred from the ITS2 reads—fungi—ranged from 15 OTUs in “LEHM 2” to 402 OTUs in Fox Skull Cave. The highest OTU diversity inferred from the partial 16S reads—bacteria—was found in Root Cave (499 OTUs), with the lowest in “LEHM 2” (220 OTUs). Despite the fact that the two Lehman Caves samples were pooled from multiple sites within the cave, with each composite sample from two distinct temporal sampling events, both generally had a lower OTU diversity when compared with wild caves (Table 2).

### 3.2. Community Composition

Photoautotrophic OTUs diversity inferred form partial 23S reads varied widely among sampling sites, (Figure 2). The two temporally distinct samples from Lehman Caves—“LEHM 1” collected in May 2019 from 10 sites within the cave; and “LEHM 2” collected from five sites within the cave in November 2019—resulted in different lampenflora communities (Figure 2). The first composite sample from Lehman Caves, “LEHM 1”, was dominated by reads from unicellular algae, and a small contribution of reads from bryophytes, cyanobacteria, and diatoms. In contrast, the second composite sample from Lehman Caves, “LEHM 2”, was dominated by reads from cyanobacteria and diatoms, with a much lower proportion of reads from unicellular algae. The algal OTUs from “LEHM 1” were recovered in a separate distinct algal clade from those that dominated the “LEHM 2” sample (Appendix A). Across all caves, “LEHM 1”, Ice, and Lower Pictograph caves were dominated by unicellular algae; cyanobacteria dominated the samples from “LEHM 2”, Catamount, Upper Pictograph, and Fox Skull caves; the Root Cave sample was dominated by bryophyte reads; and the Squirrel Springs Cave sample had the most diverse photoautotroph community (Figure 2). Diatom reads made a negligible contribution to sampling of the Ice, Catamount, Lower Pictograph, and Fox Skull caves. Reads from vascular plants only made a significant contribution in the Catamount Cave sample.

Bacterial OTUs in cave biofilms inferred from partial 16S reads comprised 16 phylum-level groups and were dominated by Actinobacteriota, Bacteroidota, and Proteobacteria (Figure 3A). Moreover, 16S OTUs were assigned to 35 class-level groups, with Actinobacteria and Alphaproteobacteria comprising a major proportion of the overall bacterial diversity in all caves (Figure 3B). Bacterial communities across caves were strikingly different at the order level, including the two Lehman Caves samples (Figure 3C).

The majority of fungal OTUs inferred from ITS2 reads were identified as Ascomycetes, although Basidiomycetes were also recovered across all samples, albeit in much lower abundance (Figure 4A). The presence of Chytridiomycetes, Mortierellomycetes, Mucoromycetes, and Rozellomycetes was also inferred, although each in low relative abundance; however, a substantial proportion of OTUs were not identifiable at any taxonomic level (Figure 4A). Within the Ascomycetes, OTUs from Dothideomycetes were most abundant in wild cave samples, and Lecanoromycetes and Eurotiomycetes were also frequently inferred from short read data (Figure 4B). The two temporally distinct composite samples from Lehman Caves resulted in vastly different Ascomycete communities, with Eurotiomycetes dominating “LEHM 1” and Sordariomycetes dominating “LEHM2”, with Eurotiomycetes reads virtually absent from the latter (Figure 4B). Furthermore, the most common fungal class in wild caves, Dothideomycetes, was found in very low abundance in “LEHM 1” and absent from the “LEHM 2” samples. In addition, reads from the generally lichen-forming class Lecanoromycetes were frequent in “LEHM 2” and wild cave samples but completely absent from the “LEHM 1” samples. Tremellomycetes were the most abundant Basidiomycete class, and Tremellomycetes OTUs were found in significant abundance in all caves (Figure 4C). Reads assigned to Agaricomycetes were common in six of the seven wild caves but absent in the lampenflora communities of Lehman Caves. Given the current expansion of *Pseudogymnoascus destructans* (Blehert & Gargas) Minnis & D.L. Lindner [51], the causative agent of white-nose disease in bats, fungal OTUs were screened for the presence of this pathogen to assess if this pathogen has reached Great Basin National Park. While *P. destructans* was not detected in any of the samples from GRBA, three OTUs were assigned to the genus *Pseudogymnoascus*—OTUs “OTU1389” (in Lehman Cave), “OTU1402” (all caves except Fox and Upper Pictograph), and “OTU1441” (all caves except Fox and Upper Pictograph). The closest match to “OTU1389” was an unidentified *Pseudogymnoascus* species, at ca. 97.9% similarity; “OTU1402” was identical to accession from various *Pseudogymnoascus* species; and “OTU1441” was most similar to *P. pannorum*, a species closely related to *P. destructans*. No attempts were made to identify other potential pathogens.

### 3.3. Community Analyses

Photoautotrophic communities in GRBA caves were generally highly dissimilar (Figure 5). Cyanobacteria, diatoms, and algae OTU diversity in Lehman Caves is only a partial subset of the composite OTU diversity occurring in wild caves (Figure 5, Venn diagrams). The majority of cyanobacteria and diatom OTUs occurring in Lehman Caves were not sampled in any of the wild caves in GRBA investigated for this study (Figure 5A,B, Vennn Diagrams). Similarly, five of the eight algal OTUs from Lehman Caves were not found in the nearby wild caves (Figure 5C, Venn diagram). In contrast, all bryophyte OTUs occurring in Lehman Caves were also found in the nearby wild caves (Figure 5D, Venn diagram). NMDS analyses corroborated the inference of highly dissimilar photoautotrophic communities among sampled caves (Figure 5, nMDS plots). The two composite Lehman Caves lampenflora samples were also shown to be distinct from the wild cave samples, with the exception of the bryophyte component, and generally more similar to each other (Figure 5).

Our sampling revealed that cyanobacterial communities in each of the eight caves sites were no more than 20% similar to each other, with the exception of Catamount and Fox Skull Caves and the two composite samples from Lehman Caves, which showed a higher percentage of similarity to each other, respectively (Figure 5A, cluster diagram). Similarly, diatom communities were highly dissimilar among caves, with the exceptions Root and Upper Pictograph caves showing 55% similarity and Ice and Catamount Cave showing 35% similarity (Figure 5B, cluster diagram). Within Lehman Cave, diatom communities were ca. 80% similar between the first samples, “LEHM 1”collected in May 2019, and the second set of samples, “LEHM 2” collected in November 2019 (Figure 5B). In general, algal communities showed a moderate degree of similarity of lampenflora components among caves, with Upper Pictograph Cave most dissimilar to the others (Figure 5C) and “LEHM 1” and “LEHM 2”with ca. 40% similarity. Fox Skull, Squirrel Springs, Root and Upper Pictograph caves showed 40% similarity in bryophyte communities, and Root and Upper Pictograph had over 60% similarity (Figure 5D).

Similar to photoautotroph communities in Lehman Caves, bacterial and fungal OTU diversity in Lehman Caves is only a partial subset of the composite OTU diversity occurring in wild caves (Figure 6). The majority of bacterial and fungal OTUs from Lehman Caves were not sampled in nearby wild caves (Figure 6, Venn diagrams). NMDS analyses of bacterial and fungal OTUs also inferred that biofilm communities associated within lampenflora in Lehman Caves were distinct from communities in wild caves (Figure 6, nMDS). Cluster analyses revealed overall similarity in bacterial communities among caves between ca. 15 and 50%, while fungal communities showed lower similarity, ranging from 0 to ca. 20% similarity (Figure 6, cluster diagrams). ANOSIM tests did not reveal significant similarities among caves within watersheds or based on the presence or absence of bats.

## 4. Discussion

Here, we used high-throughput amplicon sequencing, targeting three different loci, to provide the most comprehensive perspective to-date into the microbial and lampenflora biodiversity of a highly developed cave in comparison with wild caves in Great Basin National Park, Nevada, USA. Most sampled caves supported diverse and distinct bacterial, fungal, and lampenflora communities, with little overlap among caves (Table 2). Photoautotrophic members of these communities were generally less diverse than the bacteria or fungi but showed similar patterns of distinctive, non-overlapping communities within each cave. Overall, the microbial and lampenflora communities in the show cave—Lehman Caves—were distinct, and generally less diverse, from those occurring in nearby unlit cave entrances (Table 2; Figure 2, Figure 3 and Figure 4). These results suggest an unidentified source for a significant proportion of the lampenflora community members (including bacteria, and fungi) in Lehman Caves, with most OTUs not found in nearby wild caves. Below, we discuss the implications of our findings and directions for future work.

The presence of humans, even in caves, has a documented and substantial impact on the diversity of microbial communities, e.g., [6,9,15,20,52]. In Lehman Caves, the photosynthetic microbial communities around artificial lighting—cyanobacteria, diatoms, and algae—were distinct from communities sampled on rock walls near natural light in the wild caves (Table 2; Figure 2 and Figure 5), suggesting the potential direct impact of human visitors on lampenflora communities. Lampenflora composition around artificial lighting is likely impacted by seasonal treatment of sodium hypochlorite, NaClO (bleach), used to reduce the growth and impact lampenflora [6]. However, regardless of the temporal proximity to bleach treatments, samples in Lehman Caves were still more similar to each other than to the communities in wild caves, further supporting the perspective that members of the lampenflora communities did not originate from the walls of nearby sampled wild caves. These results suggest that human visitation may be the source of novel microbial community members. However, we also note that differences in abiotic factors and seasonal dynamics among the show cave and wild caves likely play fundamental roles in structuring biofilm communities that have not yet been characterized [21]. Due to the lack of site-specific metadata, e.g., geochemistry, temperature, relative humidity, etc., and a sampling strategy that did not assess seasonal dynamics, additional research will be required to address these questions.

The impact of periodic NaClO treatments on lampenflora communities in Lehman Caves remains unclear. Despite the observation that communities associated with lampenflora are less diverse than those in naturally lit caves, our results do not provide clear evidence that periodic NaClO treatments effectively reduced diversity in lampenflora communities. While lampenflora biomass increased during the six months post-NaClO treatments (personal observation—not quantified), we observed that the total OTU counts were reduced by almost 25% between the first (sampled in May 2019) and second (November 2019) sample periods in Lehman Caves, particularly among the green algae and fungi (Table 2). Whether the reduction in overall diversity was the result of seasonal fluctuations, succession, or other factors is not clear [2]. We note that in contrast to green algae and fungi, cyanobacteria and bacteria diversity was more stable across the two sampling periods in Lehman Caves. We also documented an increase in diatom diversity in the second temporal sample in Lehman Caves. However, we note that the two samples from Lehman Caves may not be directly comparable, with the first sample representing 10 independent artificially lit sites within Lehman Caves (May 2019) and the second sample representing 5 of the original 10 lampenflora sites within the cave (November 2019). Overall, our results corroborate the perspective that the long-term use of NaClO treatments may not be an effective means of controlling lampenflora communities [6]. However, many of the sampled sites in Lehman Caves were selected because they had especially copious amounts of lampenflora and may not be representative of lampenflora communities throughout the cave. Furthermore, we note that at these sites, NaClO treatments temporarily reduce the visible concentration of lampenflora. Future research investigating the temporal and spatial dynamics of lampenflora communities, and their relationship to periodic cleaning treatments, will be necessary to determine the best practices for controlling lampenflora growth [5,9,15,53]. Although several physical, chemical, and biological methods can be adopted to help control lampenflora, reducing the intensity and period of illumination is the most effective strategy for controlling growth [53].

Other environmental factors might also impact microbial and lampenflora diversity in caves. A year-long longitudinal study of microbial succession in a karst cave system in Virginia, USA, showed that large influxes of aquatic microbes and particulate organic matter can enter the cave from either the surface or interstitial zones [54]. This results in aquatic bacterioplankton communities within caves that are highly similar to those occurring near the surface when there is a direct hydrologic connection [54]. In contrast, the majority of microbes on rock walls, speleothems, and cave soils often form endemic communities [22,25,55]. However, when humans enter caves, biological communities are consistently altered [56,57,58]. Our results from Great Basin National Park were in line with the perspective that rock walls and speleothems from different caves support distinctive microbial communities. However, the range of variation in community composition among distinct lampenflora communities (including bacteria and fungi) in Lehman Caves has not been characterized, although distinct communities have been documented in different habitats within the cave [19]. Airborne allochthonous material or other air microflora might represent an important and continuous source of organic material for cave fauna [7], and carefully designed research investigating potential lampenflora sources in Lehman Caves relative to broad patterns of air movement may illucidate important patterns. By more fully characterizing these communities, we can better monitor the establishment of lampenflora and inform effective strategies for their management and removal. Future alternatives to bleach solutions may include the use of biocides that degrade key structural components of the biofilm to decrease biomass or use of the NaClO solution first and then a targeted biocide treatment to inhibit reestablishment [59]. However, before more targeted approaches can be adopted, research must be done to identify which clades of lampenflora determine the biofilm structure in the caves. For example, another avenue yet to be explored is using a treatment system that targets the extracellular polysaccharides that hold the biofilm together and to the cave, as well as that are responsible for storing water when humidity decreases [60].

In caves in Great Basin National Park, bacteria—inferred from partial 16S reads—were generally the most diverse group sampled, followed by fungi—inferred from ITS2 reads, and photosynthesizing organisms—inferred from partial 23S reads (Table 2). Within bacteria, *Proteobacteria* or *Actinobacteria* were the most dominant lineages recorded in each cave. In other cave systems, some *Actinobacteria* have been shown to capture CO_2_ to dissolve the rock and subsequently generate CaCO_3_ in periods of lower humidity and/or CO_2_ [61], potentially impacting speleothem deposition [62]. However, *Proteobacteria* and *Actinobacteria* have been noted in the Manao-Pee cave system (Thailand) without environmental damage and in similar abundances as found in Lehman Caves [24]. *Actinobacteria* in lower abundances may promote early speleothem deposition [63]. This would indicate that treatments should not be done to purge biofilm completely from cave systems, but to manage lampenflora components so they do not overgrow. Therefore, further treatments may focus on decreasing the light intensity the lampenflora receive, instead of removing them all together. Experimental studies of cultured bacteria from Lehman Caves may provide insight into which species are responding most strongly to the artificial lighting and why *Proteobacteria* and/or *Actinobacteria* are dominant in these caves. Fungal communities were also relatively diverse in most sampled caves—between 15 to 402 fungal OTUs/cave, with the lowest diversity in Lehman Caves and Upper Pictograph Cave (Table 2). Fungal communities were dominated by *Ascomycetes* and substantial proportions of unidentified fungal OTUs (Figure 4). We note that there was no evidence from fungal ITS2 amplicons, suggesting the presence of *Pseudogymnoascus desctructans*—the psychrophilic fungus that causes white-nose syndrome in bats [64]—in Great Basin National Park. In general, photosynthesizing organisms were the least diverse group sampled in caves in Great Basin National Park, even in the lampenflora communities around artificial lighting in Lehman Caves (Table 2). However, metabarcoding also revealed that Lehman Caves had more diverse cyanobacteria communities than the wild caves. We note that while this study found over 150 cyanobacterial OTUs, pristine, unlit caves may typically harbor low frequency of cyanobacteria [65].

In conclusion, our study of microbial and lampenflora communities on rock surfaces in caves in Great Basin National Park reveals that these harbor diverse and distinct communities. Our data did not support the hypothesis that members of lampenflora communities in Lehman Caves were a subset of communities from rock surfaces in other, nearby wild caves. Rather, our results indicate that members of lampenflora communities (including bacteria and fungi) originate from presently unidentified sources. Our results provide a valuable framework for future work exploring lampenflora community assembly in both show and wild caves in Great Basin National Park, including a general perspective on the range of microbial and lampenflora species occurring on rock surfaces in caves in the Great Basin.

## Figures and Tables

**Figure 1 microorganisms-09-01188-f001:**
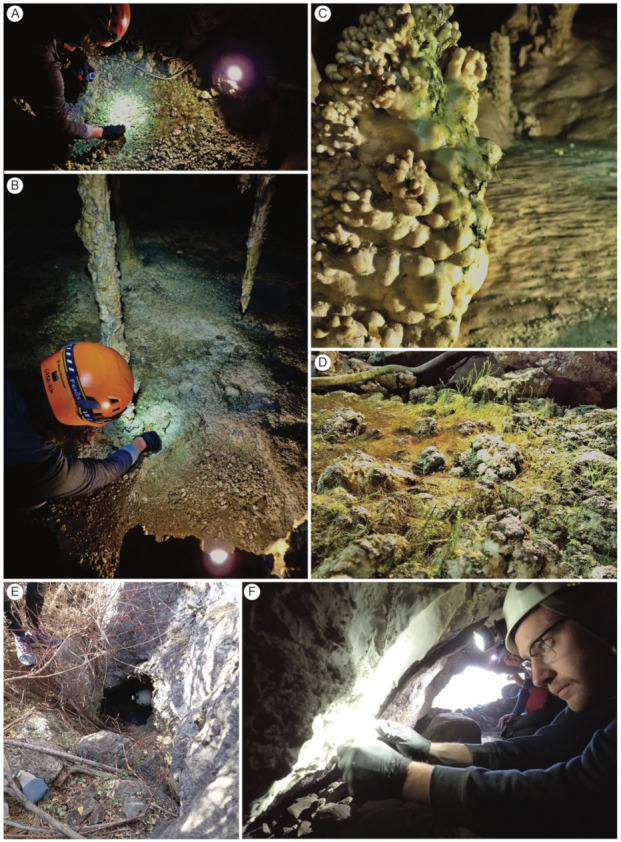
Examples of cave sample sites in Great Basin National Park. (**A**) lampenflora community on cobble fill of cave floor in Lehman Caves. (**B**) Distribution of lampenflora community around artificial lighting in Lehman Caves. (**C**) Lampenflora community and chlorophyll staining on stalagmite in Lehman Caves. (**D**) Bryophyte-dominated lampenflora community in Lehman Caves. (**E**) Sampling in Squirrel Springs Cave in the Snake Creek watershed. (**F**) Sampling in Upper Pictograph Cave.

**Figure 2 microorganisms-09-01188-f002:**
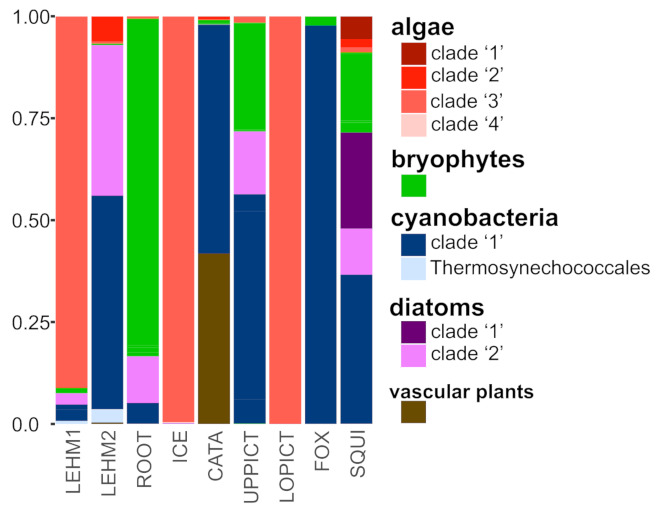
Overview of photoautotroph diversity at eight caves in Great Basin National Park inferred from environmental DNA metabarcoding of partial 23S amplicons. OTUs were broadly assigned to one of five broad groups—“algae”, “bryophytes”, “cyanobacteria”, “diatoms”, and “vascular plants” based on phylogenetic reconstructions of the alignment of the 23S fragment and BLAST comparisons. Bar plot depicts the relative proportion of reads assigned to cyanobacteria, algae, bryophytes, diatoms, and vascular plants—each column represents a sampled cave (see Table 1). In “algae”, “cyanobacteria”, and “diatoms”, additional groupings were reported that coincided with well-supported monophyletic clades. The proportion of reads assigned to each group is reported for each cave. The show cave with artificial lighting, Lehman Cave, was represented by two composite samples: “LEHM 1” collected in May 2019 from 10 sites within the cave; and “LEHM 2” collected from five sites within the cave in November 2019.

**Figure 3 microorganisms-09-01188-f003:**
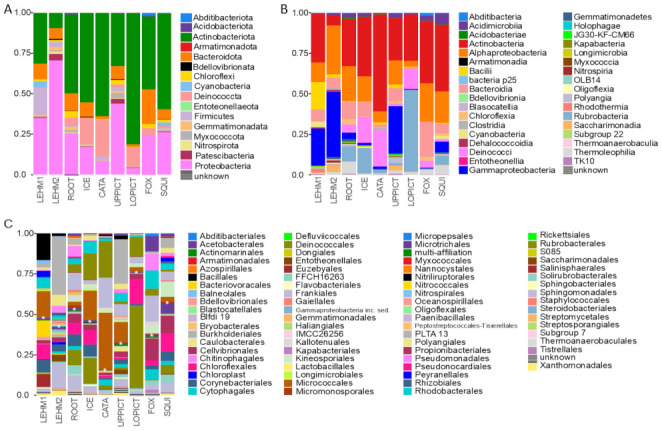
Overview of bacterial diversity at eight caves in Great Basin National Park inferred from environmental DNA metabarcoding of partial 16S amplicons. Bar plot depicts the relative proportion of reads assigned to different taxonomic ranks—each column represents a sampled cave (see Table 1). (**A**) The proportion of 16S reads assigned to phylum-level groups, inferred using the FROGS pipeline, for each cave. (**B**) The proportion of 16S reads assigned to class-level groups, inferred using the FROGS pipeline, for each cave. (**C**) The proportion of 16S reads assigned to order-level groups, inferred using the FROGS pipeline, for each cave. The show cave with artificial lighting, Lehman Caves, was represented by two composite samples: “LEHM 1” collected in May 2019 from 10 sites within the cave; and ‘LEHM 2′ collected from five sites within the cave in November 2019. Note: in panel (**C**), the color pattern repeats starting with the order “Micropepsales” and the repeat is indicated with a white asterisk in each stacked bar plot.

**Figure 4 microorganisms-09-01188-f004:**
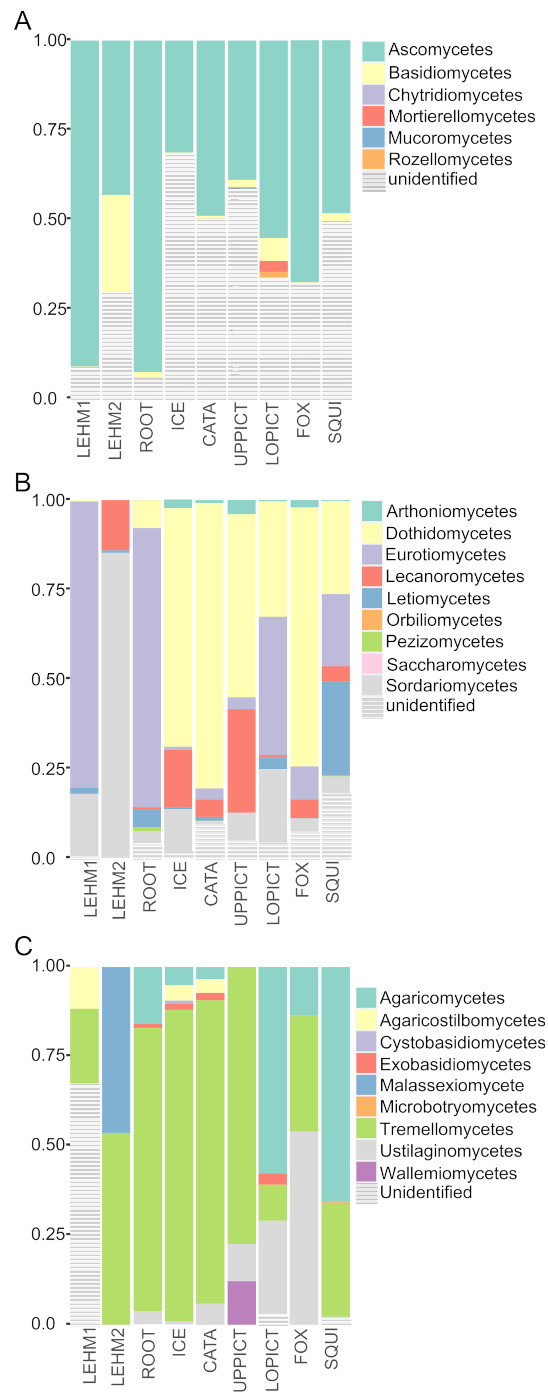
Overview of fungal diversity at eight caves in Great Basin National Park inferred from environmental DNA metabarcoding of ITS amplicons. Bar plot depicts the relative proportion of reads assigned to different taxonomic ranks—each column represents a sampled cave (see Table 1). (**A**) The proportion of ITS reads assigned to phylum-level groups, inferred using the PIPITS pipeline, for each cave. (**B**) Class-level assignments for ascomycete OTUs; all non-ascomycete reads are excluded (see panel (**A**)), and the relative abundance of remaining ascomycete classes were adjusted proportionally. (**C**) Class-level assignments for basidiomycete OTUs; all non-basidiomycete reads are excluded (see panel (**A**)), and the relative abundance of remaining basidiomycete classes were adjusted proportionally.

**Figure 5 microorganisms-09-01188-f005:**
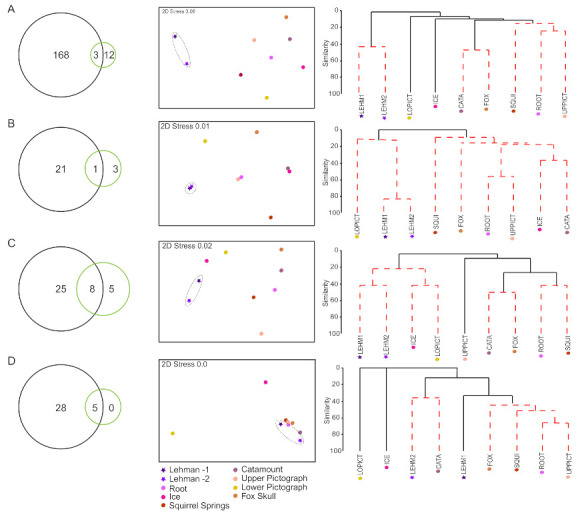
Comparison of photoautotroph communities in Lehman Caves (show cave) with wild caves: (**A**) cyanobacteria; (**B**) diatoms; (**C**) algae; and (**D**) bryophytes. In the Venn diagrams on the left-hand side of each panel, OTUs from the Lehman Caves samples (“LEHM1” and “LEHM2”) were combined and are shown with green circles; similarly, OTUs from the seven wild caves were combined into a single composite “wild caves” sample and are shown in black circles. NMDS plots are shown in the center of each panel. Cluster diagrams are on right-hand side of each panel, with red dashed lines indicating statistically different groups. The show cave with artificial lighting, Lehman Cave, was represented by two composite samples: “LEHM 1” collected in May 2019 from 10 sites within the cave; and “LEHM 2” collected from five sites within the cave in November 2019. See Table 1 for additional details on sampling sites.

**Figure 6 microorganisms-09-01188-f006:**
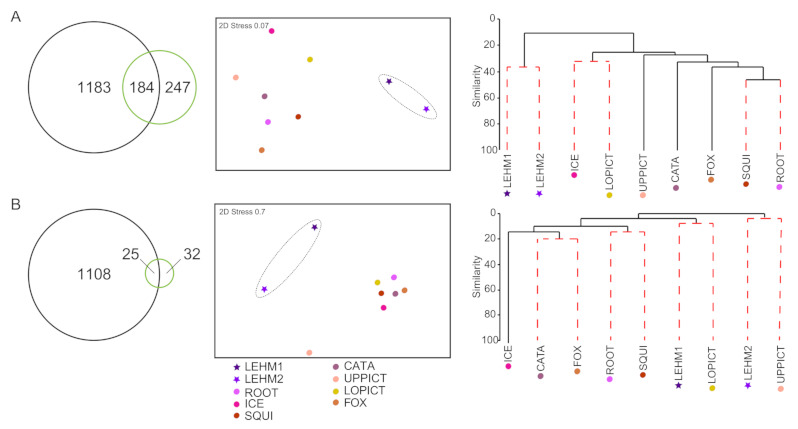
Comparison of bacterial (**A**) and fungal communities (**B**) in Lehman Caves (show cave) with wild caves. In the Venn diagrams on the left-hand side of each panel, OTUs from the Lehman Caves samples (“LEHM 1” and “LEHM 2”) were combined and are shown with green circles; similarly, OTUs from the seven wild caves were combined into a single composite “wild caves” sample and are shown in black circles. NMDS plots are shown in the center of each panel. Cluster diagrams are on the right-hand side of each panel, with red dashed lines indicating statistically different groups. The show cave with artificial lighting, Lehman Cave, was represented by two composite samples: “LEHM 1” collected in May 2019 from 10 sites within the cave; and “LEHM 2” collected from five sites within the cave in November 2019.

**Table 1 microorganisms-09-01188-t001:** Overview of caves in Great Basin National Park (GRBA) sampled for this study. Biofilm samples in GRBA were collected from naturally lit marble, limestone, and dolomite surfaces within (near the entrances) eight “wild” caves—caves with infrequent human visitation and without artificial lighting—distributed across three drainages, in addition to artificially lit lampenflora communities in Lehman Caves (show cave).

Cave Name/Sample Date	Watershed	Length	Depth	Elevation (Meters above Sea Level)	Water Near Sampling Sites	Notes
Catamount (“CATA”)—wild (Pole Canyon Limestone)/15 November 2019	Baker	~10 m.	~9 m.	~2179 m.a.s.l.	no	Short, enlarged fracture cave
Ice (“ICE”)—wild (Pole Canyon Limestone)/31 May 2019	Baker	206 m.	12 m.	2159 m.a.s.l.	yes	Occasionally has ice near entrance (temperatures about 10 °F lower than other caves)
Upper Pictograph (“UPPICT”)—wild (Pole Canyon Limestone)/15 November 2019	Baker	56 m.	6 m.	2179 m.a.s.l.	no	Pictographs and bats (five species); dust from adjacent gravel road
Lower Pictograph (“LOPICT”)—wild (Pole Canyon Limestone)/15 November 2019	Baker	49 m.	5 m.	2163 m.a.s.l.	no	Pictographs; dust from adjacent gravel road
Lehman (“LEHM”)—show (Pole Canyon Limestone)/31 May & 15 November 2019	Lehman	~3353 m.	~30 m.	2109 m.a.s.l.	yes	Longest cave in Nevada
Root (“ROOT”)—wild (Pole Canyon Limestone)/15 November 2019	Lehman	56 m.	9 m.	2105 m.a.s.l.	no	Steep, narrow entrance
Fox Skull (“FOX”)—wild (Notch Peak Limestone)/15 November 2019	Snake	31 m.	3 m.	1998 m.a.s.l.	no	Attracts a lot of wildlife
Snake Creek (“SNAKE”)—wild (Notch Peak Limestone)	Snake	513 m.	17 m.	1997 m.a.s.l.	no	Varied passages and speleothems
Squirrel Springs (“SQUI”)—wild (Fish Haven and Laketown dolomites)	Snake	16 m.	7 m.	2200 m.a.s.l.	yes	Sumps during wet years

**Table 2 microorganisms-09-01188-t002:** Summary of amplicon-sequencing diversity at lit surfaces in eight caves in Great Basin National Park. The number Illumina reads are reported for each cave—partial 16S amplicons targeting bacteria, partial 23S amplicons targeting photoautotrophs, and ITS2 amplicons targeting fungi. The number of operational taxonomic units (OTU) represented by >4 reads/OTU are reported for major organismal groups in each cave. The total number of “lampenflora” (cyanobacteria + green algae + bryophytes) OTUs is summarized for each cave, as is the total number of OTUs from each cave.

	Lehman 1	Lehman 2	Root	Ice	Catamount	Upper Pictograph	Lower Pictograph	Squirrel	Fox
16S reads	83.9k	38.7k	115.2k	106.5k	109.5k	107.3k	112.1k	91.4k	240.4k
23S reads	97.5k	104.9k	117.5k	95.6k	131.1k	99.5k	110.0k	105.7k	114.8k
ITS2 reads	85.5k	77.2k	109.0k	107.3k	77.9k	123.6k	70.0k	124.7k	88.5k
Cyanobacteria (23S) OTUs	12	12	26	1	107	32	0	31	103
Green algae (23S) OTUs	13	2	10	14	6	2	10	15	1
Diatoms (23S) OTUs	3	4	7	1	2	6	0	16	0
Bryophytes (23S) OTUs	5	1	21	0	5	13	0	18	3
Lampenflora Total OTUs	33	19	64	16	120	53	10	80	107
Fungi (ITS2) OTUs	48	15	182	136	336	33	255	280	402
Bacteria (16S) OTUs	286	220	499	234	279	523	237	426	477
Total OTUs	367	254	745	386	735	609	502	786	986

## Data Availability

Short reads generated for this study were deposited in NCBI’s Short Read Archive under BioProject accession number PRJNA731535.

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
