# Peer review of "Lampenflora in a Show Cave in the Great Basin Is Distinct from Communities on Naturally Lit Rock Surfaces in Nearby Wild Caves"

_microorganisms, 2021, doi:10.3390/microorganisms9061188_

Round 1

Reviewer 1 Report

Dear Authors

the paper is an important addition to the numerous studies done on lampenflora worldwide. My observations are in the attached document and below:

  • there are important papers for lampenflora and human impact on microorganisms in caves which were ignored
  • there are many repetitions in the text, please try to avoid them 
  • your conclusions are too general. They should be narrowed to your samples and context. Caves are complex habitats, with numerous micro-habitats, extremely different especially on microorganisms. Cave microorganisms are endemic, while microorganisms in show caves are dominated by allochthonous taxa
  • be careful with the spelling errors, they are many in the text.

Reviewer 2 Report

The manuscript ID microorganisms-1215529, entitled `Who finds the light? Searching for the source of lampenflora communities in show caves` by Burgoyne et al. describes the microbial communities associated to artificially and naturally-lightened sites in eight karst caves. The results are novel and of interest for the understanding of human-induced changes in the trophic webs of cave systems. By using amplicon-sequencing approach, authors outlined that microbial communities detected in the Lehman cave samples that were exposed to artificial light are substantially different from those analysed in the photic-aphotic transition zones of other `wild` caves. The hypothesis and aim of research are clear-cud defined and methodology although not diverse, well applied.

Major concern

In my opinion, the report would be much improved if the amplicon-sequencing results (i.e. community structure and composition) would have been integrated by metadata such as geochemical data of sampled rocks. These variables might explain or rule out the reasons for the differences found among the samples.

Minor concerns:

  • the title should be more informative. I see the choice of authors for a catchy title but this does not look too scientific and it rather appear as popular science communication and discloses too little on the contents of the manuscript. Additionally, in the way that the title is formulated now, one would expect that the paper will give a clear answer for `the source of lampenflora` investigated, which is not the case. I suggest a straightforward scientific (say, conventional) title.
  • Adding country name (USA) to the affiliation of authors is adviseable
  • A very important note: `metagenomc(s)` term should not be used at all throughout the manuscript. The approach used IS NOT metagenomics (that is the whole-community shotgun sequencing which is NOT THE CASE here). Please delete this word that may confuse the readers and it is, however, inappropriately used. Instead, the approach is `amplicon-sequencing` (that is, the sequencing of selected taxonomy marker gene only). Alternatively, `environmental DNA metabarcoding` term could be used. Therefore, I strongly suggest removing `metagenomics(s)` throughout the text and figure captions and replacing it with the terms suggested above.
  • Throughout the manuscript I recommend the authors to avoid mixing the description of bryophytes as part of microbial communities. The report is focused on amplicon-sequencing-based diversity of microorganisms but bryophytes are mentioned among these which is misleading for the readers. Carefully explain and describe the bryophytes as separate issue (for example, as the fungi are described). Explain why bryophyte sequences (probably the plastid 23S rRNA genes were amplified during total PCR amplification).
  • Throughout the manuscript, I recommend using `(partial) 16S or 23S rRNA gene (fragments)` (or a combination of this wording) instead of simply 16S/23S or `rDNA`.
  • Ln 19 –the listing of organisms should start with prokaryotes and ended with eukaryotes (bryophytes)
  • Ln 35 -`assemblies` instead of `assembly`
  • Ln 37 `establishment OF lampenflora`
  • Ln 53 `which` instead of `with`
  • Ln 58 –delete IN within `niche in provides`
  • Ln 62 –the statement here is odd `novel ecological networks ….may pose serious problems in cave communities [6]`. I feel that the message is unclear. The microbial communities (if am I right) simply don’t care about changing of developing of novel ecological networks, they simply adapt and change. Something is not clear in the statement, anyway.
  • Ln 66 –the statement that the `methods developed by the medical microbiology` is misleading or not appropriate. Why mention this? Is this relevant in the context of the paper? Traditional microbiological methods (microscopy/ cultivation) were also developed as tools in industrial microbiology as well.
  • Ln 114 – please write once again the full name of GRBA and specify the location (western USA)
  • Ln 124 – It would be useful if a reference reporting the bleaching procedure in Lehman Cave is cited here
  • It would also be useful if the light intensity (in lumens) for the LEDs used Lehman Cave is given here and also in the Ln 137 following the wording `artificially lit…`. This information might be relevant for data comparison by other studies.
  • Table 1 – I suggest adding GPS coordinates for all tested caves.
  • As the description of sampling/ pooling strategy for Lehman Cave looks somehow difficult to comprehend (lines 125-128), I suggest a clearer description of this sampling in Lehman supported by an extra column in Table 1 indicating the precise sampling point and number of samplings for each cave. In this way, the reader could much easier visualize the sampling strategy by looking at the emended table 1.
  • Table 1 – in addition to the above suggestions, I would strongly advise to add the sampling times (day/month/year).
  • Lines 120 and 129-130 comprise repetitive information. Please describe the sample preservation in one sentence for all cases.
  • The y-axes in Figures 2 to 4 should be explained - titles added directly in the plots.
  • Table 1 – I recommend using the metric system (for length/depth/elevation). In the actual form, the table include data expressed in feet which is not conventional in the literature.
  • Ln 141 and elsewhere – to the brand name (Zymo Research, Life Technologies, etc.) please add city/country as this is quite a conventional way of indicating the manufacturers `(brand name, city, country)`. Please be consistent throughout the manuscript.
  • Ln 147 –indicating `plastid 23S rRNA gene` as present in cyanobacteria is scientifically incorrect as cyanobacteria do not hav plastids. My advice is to re-formulate this by mentioning that this gene is present in cyanobacteria and in the plastids of algae.
  • 156 –Qubit
  • Ln 161 – it is essential to delineate the cut-off limit for OUT clustering. What is this limit 0.03% difference (97% similarity) is the widely accepted standard. Did authors use this limit, they should write this. Have they used another cut-off value (0.05 / 95%?), they should specify it. All cut-off values used for clustering the OTUs from 23S/ 16S rRNA gene fragment and ITS2 reads must be exactly mentioned.
  • Ln 194 – I suggest `normalized` instead of `standardized`. If the meaning intended by authors is for `standardized`, please explain.
  • Ln 208 and elsewhere –metagenomics was not employed. As suggested above, the name of the approach should be corrected.
  • Ln 210 – add am accession code or an ID to the project name PENDING
  • Lns 219, 231 and throughout the manuscript I found typos of the name of `Lehman` - please correct them
  • Table 2 – the caption is meant to describe the `microbial diversity` although bryophytes and basidiomycetes are not microbes. One suggestion is `Amplicon-sequencing-based diversity…` or alternate wording.
  • Figure 2 – I am not algologist nor familiar with algae taxonomy but I expected a more informative denominations for algal diversity. Delineating Clades 1-4 is very vague to me. Same for `vascular plants` and diaBLU (is this an accepted taxonomic name?). Same for chromophytic algae (Chromista?). I suggest correcting these quite generic taxon names toward a more scientific and informative names.
  • In the same topic, I have noticed the detection of `diaPINK` diatoms listed in the Supplementary files (23s_ALGAE…) but this group is apparently not present in the Figure 2.
  • In Figure 2 –is still unclear what Clade 1/ Cyanobacteria include. When checking the Supplementary file `23s_ALGAE-` I have seen that cyanobacteria from Nostocales, Synechococcales and Phormdiales were detected in addition to Thermosynechococcales. It is feasible to detail the diversity of cyanobacteria in the Figure 2?
  • Lns 290-300 – I am used to capital and italic letters for taxa. In this paragraph most of the fungal taxonomy is written with lowercase/regular letters. Please be consistent in writing taxa names throughout the manuscript. At lns 497-498, the taxa names are written in italics.
  • Ln 291 - `across all`
  • Ln 299 - `virtually` (?)
  • Ln 306 –what is the rationale for Pseudogymnoascus destructans screening? Is there a standard in screening for this pathogen during fungi assessment in caves or this is motivated by the presence of bat colonies across the tested caves?
  • Throughout Figure captions I recommend avoiding repetitive description of samples. Once described (for example in Materials and Methods, or in the caption of Figure 2), the information in other captions could be omitted.
  • Ln 369-379 – some text marks were left on the Figure references
  • Ln 372 -`occurring`
  • NMDS abbreviation was mentioned with uppercase N in Ln 198. The convention is that NMDS should be written with uppercase N but at Ln 376 onward, nMDS (lowercase n) is used. Be consistent and use NMDS.
  • Ln 428 – the statement `DNA-based perspective` is vague and too general. Please be more specific.
  • Ln 451-452 – In my view no thorough analysis of abiotic factor-dependent changes in diversity was made. No seasonal dynamics could be inferred as well as the investigation did not evaluate the diversity in multiple time point or seasons. Detailed data on abiotic factors are missing (see my main concern).
  • Ln 458 – same conceptual problem as above: no `lampenflora biomass` assessment has been performed (i.e. biomass quantification). Therefore, such statements lack evidences. As far as I could understand, only cave-dependent changes in structure and composition and abundance of gene marker reads was performed.
  • Lns 464-466 – the statement ending with `reduced competition from the green algae and fungi` is highly speculative if not supported by direct evidences or literature.
  • Ln 489 –what is `biological communities`?
  • Ln 505 –what is `genetic evidence`? If evidences from amplicon-sequencing are meant, I doubt that the word `genetic` is appropriately used here. Please re-word.
  • Ln 519 – the wording `Into the range of potential community members` is not clear to me. Perhaps another clearer statement would improve the message.

The manuscript ID microorganisms-1215529, entitled `Who finds the light? Searching for the source of lampenflora communities in show caves` by Burgoyne et al. describes the microbial communities associated to artificially and naturally-lightened sites in eight karst caves. The results are novel and of interest for the understanding of human-induced changes in the trophic webs of cave systems. By using amplicon-sequencing approach, authors outlined that microbial communities detected in the Lehman cave samples that were exposed to artificial light are substantially different from those analysed in the photic-aphotic transition zones of other `wild` caves. The hypothesis and aim of research are clear-cud defined and methodology although not diverse, well applied.

Major concern

In my opinion, the report would be much improved if the amplicon-sequencing results (i.e. community structure and composition) would have been integrated by metadata such as geochemical data of sampled rocks. These variables might explain or rule out the reasons for the differences found among the samples.

Minor concerns:

  • the title should be more informative. I see the choice of authors for a catchy title but this does not look too scientific and it rather appear as popular science communication and discloses too little on the contents of the manuscript. Additionally, in the way that the title is formulated now, one would expect that the paper will give a clear answer for `the source of lampenflora` investigated, which is not the case. I suggest a straightforward scientific (say, conventional) title.
  • Adding country name (USA) to the affiliation of authors is adviseable
  • A very important note: `metagenomc(s)` term should not be used at all throughout the manuscript. The approach used IS NOT metagenomics (that is the whole-community shotgun sequencing which is NOT THE CASE here). Please delete this word that may confuse the readers and it is, however, inappropriately used. Instead, the approach is `amplicon-sequencing` (that is, the sequencing of selected taxonomy marker gene only). Alternatively, `environmental DNA metabarcoding` term could be used. Therefore, I strongly suggest removing `metagenomics(s)` throughout the text and figure captions and replacing it with the terms suggested above.
  • Throughout the manuscript I recommend the authors to avoid mixing the description of bryophytes as part of microbial communities. The report is focused on amplicon-sequencing-based diversity of microorganisms but bryophytes are mentioned among these which is misleading for the readers. Carefully explain and describe the bryophytes as separate issue (for example, as the fungi are described). Explain why bryophyte sequences (probably the plastid 23S rRNA genes were amplified during total PCR amplification).
  • Throughout the manuscript, I recommend using `(partial) 16S or 23S rRNA gene (fragments)` (or a combination of this wording) instead of simply 16S/23S or `rDNA`.
  • Ln 19 –the listing of organisms should start with prokaryotes and ended with eukaryotes (bryophytes)
  • Ln 35 -`assemblies` instead of `assembly`
  • Ln 37 `establishment OF lampenflora`
  • Ln 53 `which` instead of `with`
  • Ln 58 –delete IN within `niche in provides`
  • Ln 62 –the statement here is odd `novel ecological networks ….may pose serious problems in cave communities [6]`. I feel that the message is unclear. The microbial communities (if am I right) simply don’t care about changing of developing of novel ecological networks, they simply adapt and change. Something is not clear in the statement, anyway.
  • Ln 66 –the statement that the `methods developed by the medical microbiology` is misleading or not appropriate. Why mention this? Is this relevant in the context of the paper? Traditional microbiological methods (microscopy/ cultivation) were also developed as tools in industrial microbiology as well.
  • Ln 114 – please write once again the full name of GRBA and specify the location (western USA)
  • Ln 124 – It would be useful if a reference reporting the bleaching procedure in Lehman Cave is cited here
  • It would also be useful if the light intensity (in lumens) for the LEDs used Lehman Cave is given here and also in the Ln 137 following the wording `artificially lit…`. This information might be relevant for data comparison by other studies.
  • Table 1 – I suggest adding GPS coordinates for all tested caves.
  • As the description of sampling/ pooling strategy for Lehman Cave looks somehow difficult to comprehend (lines 125-128), I suggest a clearer description of this sampling in Lehman supported by an extra column in Table 1 indicating the precise sampling point and number of samplings for each cave. In this way, the reader could much easier visualize the sampling strategy by looking at the emended table 1.
  • Table 1 – in addition to the above suggestions, I would strongly advise to add the sampling times (day/month/year).
  • Lines 120 and 129-130 comprise repetitive information. Please describe the sample preservation in one sentence for all cases.
  • The y-axes in Figures 2 to 4 should be explained - titles added directly in the plots.
  • Table 1 – I recommend using the metric system (for length/depth/elevation). In the actual form, the table include data expressed in feet which is not conventional in the literature.
  • Ln 141 and elsewhere – to the brand name (Zymo Research, Life Technologies, etc.) please add city/country as this is quite a conventional way of indicating the manufacturers `(brand name, city, country)`. Please be consistent throughout the manuscript.
  • Ln 147 –indicating `plastid 23S rRNA gene` as present in cyanobacteria is scientifically incorrect as cyanobacteria do not hav plastids. My advice is to re-formulate this by mentioning that this gene is present in cyanobacteria and in the plastids of algae.
  • 156 –Qubit
  • Ln 161 – it is essential to delineate the cut-off limit for OUT clustering. What is this limit 0.03% difference (97% similarity) is the widely accepted standard. Did authors use this limit, they should write this. Have they used another cut-off value (0.05 / 95%?), they should specify it. All cut-off values used for clustering the OTUs from 23S/ 16S rRNA gene fragment and ITS2 reads must be exactly mentioned.
  • Ln 194 – I suggest `normalized` instead of `standardized`. If the meaning intended by authors is for `standardized`, please explain.
  • Ln 208 and elsewhere –metagenomics was not employed. As suggested above, the name of the approach should be corrected.
  • Ln 210 – add am accession code or an ID to the project name PENDING
  • Lns 219, 231 and throughout the manuscript I found typos of the name of `Lehman` - please correct them
  • Table 2 – the caption is meant to describe the `microbial diversity` although bryophytes and basidiomycetes are not microbes. One suggestion is `Amplicon-sequencing-based diversity…` or alternate wording.
  • Figure 2 – I am not algologist nor familiar with algae taxonomy but I expected a more informative denominations for algal diversity. Delineating Clades 1-4 is very vague to me. Same for `vascular plants` and diaBLU (is this an accepted taxonomic name?). Same for chromophytic algae (Chromista?). I suggest correcting these quite generic taxon names toward a more scientific and informative names.
  • In the same topic, I have noticed the detection of `diaPINK` diatoms listed in the Supplementary files (23s_ALGAE…) but this group is apparently not present in the Figure 2.
  • In Figure 2 –is still unclear what Clade 1/ Cyanobacteria include. When checking the Supplementary file `23s_ALGAE-` I have seen that cyanobacteria from Nostocales, Synechococcales and Phormdiales were detected in addition to Thermosynechococcales. It is feasible to detail the diversity of cyanobacteria in the Figure 2?
  • Lns 290-300 – I am used to capital and italic letters for taxa. In this paragraph most of the fungal taxonomy is written with lowercase/regular letters. Please be consistent in writing taxa names throughout the manuscript. At lns 497-498, the taxa names are written in italics.
  • Ln 291 - `across all`
  • Ln 299 - `virtually` (?)
  • Ln 306 –what is the rationale for Pseudogymnoascus destructans screening? Is there a standard in screening for this pathogen during fungi assessment in caves or this is motivated by the presence of bat colonies across the tested caves?
  • Throughout Figure captions I recommend avoiding repetitive description of samples. Once described (for example in Materials and Methods, or in the caption of Figure 2), the information in other captions could be omitted.
  • Ln 369-379 – some text marks were left on the Figure references
  • Ln 372 -`occurring`
  • NMDS abbreviation was mentioned with uppercase N in Ln 198. The convention is that NMDS should be written with uppercase N but at Ln 376 onward, nMDS (lowercase n) is used. Be consistent and use NMDS.
  • Ln 428 – the statement `DNA-based perspective` is vague and too general. Please be more specific.
  • Ln 451-452 – In my view no thorough analysis of abiotic factor-dependent changes in diversity was made. No seasonal dynamics could be inferred as well as the investigation did not evaluate the diversity in multiple time point or seasons. Detailed data on abiotic factors are missing (see my main concern).
  • Ln 458 – same conceptual problem as above: no `lampenflora biomass` assessment has been performed (i.e. biomass quantification). Therefore, such statements lack evidences. As far as I could understand, only cave-dependent changes in structure and composition and abundance of gene marker reads was performed.
  • Lns 464-466 – the statement ending with `reduced competition from the green algae and fungi` is highly speculative if not supported by direct evidences or literature.
  • Ln 489 –what is `biological communities`?
  • Ln 505 –what is `genetic evidence`? If evidences from amplicon-sequencing are meant, I doubt that the word `genetic` is appropriately used here. Please re-word.
  • Ln 519 – the wording `Into the range of potential community members` is not clear to me. Perhaps another clearer statement would improve the message.
